# A comparative systematic review and meta-analysis of uterine artery resistance in pregnant women with and without previous history of cesarean section

Arash Mohazzab[1], Azar Mohammadzadeh[2], Banafsheh Nikfar [3], Safdar Masoumi[4], Elham Shoraka[5,6], Neda Hashemi[7*], Shahla Chaichian [3*]

**1** Department of Epidemiology, School of Public Health, Iran University of Medical sciences, Tehran, Iran, **2** School of Dentistry, Iran University of Medical sciences, Tehran, Iran, **3** Pars Advanced and Minimally Invasive Medical Manners Research Center, Pars Hospital, Iran University of Medical Sciences, Tehran, Iran, **4** Department of Biostatistics, Faculty of Medical Sciences, Tarbiat Modares University, Tehran, Iran, **5** School of Medicine, Islamic Azad University, Kazeroun branch, Kazeroun, Iran, **6** Department of Obstetrics and Gynecology, School of Medicine, Laparoscopy Research Center, Shiraz University of Medical Sciences, Shiraz, Iran, **7** Endometriosis Research Center, Iran University of Medical Sciences, Tehran, Iran

* nedahashemi1363@yahoo.com (NH); shchaichian@gmail.com (SC)

## Abstract

### Background

Increased uterine artery resistance in pregnant women with a history of cesarean section has been suggested to contribute to adverse pregnancy outcomes. However, the literature presents conflicting reports on this association. In this comparative meta-analysis and systematic review, we aimed to evaluate the studies that reported uterine artery resistance using Color Doppler ultrasonography in pregnant women with and without a history of cesarean section.

### Methods

We searched PubMed, Scopus, Web of Science, and Embase up to April 2024 using relevant keywords. Study selection was performed by two independent researchers, with conflicts resolved by a third. Risk of bias was assessed using the Newcastle-Ottawa Scale. The primary outcomes were the Pulsatility Index (PI) and Resistance Index (RI) Meta-analysis and meta-regression were conducted using STATA version 17.

### Results

After screening 442 articles, the meta-analysis included six studies, encompassing 1,656 participants. We found a small but statistically significant increase in uterine artery resistance, based on PI, in women with a history of cesarean section

**Data availability statement:** All relevant data are within the manuscript and its Supporting Information files.

**Funding:** The author(s) received no specific funding for this work.

**Competing interests:** The authors have declared that no competing interests exist.

(Hedges's g = 0.15, 95% CI [0.03, 0.26], p = 0.01). Heterogeneity among studies was low (I² = 26.60%, p = 0.23), and no significant publication bias was detected (Egger's test, p = 0.81). Analysis of the RI, based on two studies, showed a non-significant increase in the cesarean group (Hedges's g = 0.19, 95% CI [−0.06, 0.43], p = 0.13).

## Conclusion

A history of cesarean section may be associated with increased uterine artery resistance. These findings suggest a possible benefit in monitoring uterine artery resistance in subsequent pregnancies, mainly using Color Doppler ultrasonography, to better understand potential risks such as preeclampsia and intrauterine growth restriction. However, given the limited evidence, further studies are warranted to confirm these associations and clarify their clinical relevance.

## Introduction

Cesarean section rates have been increasing globally, resulting in a rising number of women with uterine scars from previous cesarean deliveries [1]. The necessity of assessing complications and side effects of cesarean sections is crucial due to the significant impact they can have on maternal and fetal health. Cesarean sections can lead to several short-term complications, such as infection, hemorrhage, and anesthesia-related issues, as well as long-term complications affecting subsequent pregnancies, including uterine rupture, abnormal placentation, and increased risk of preterm birth [2–4]. These complications not only pose risks to maternal health but can also result in adverse neonatal outcomes such as low birth weight and respiratory distress [5,6].

However, the literature presents conflicting reports on this association. Some studies have found a positive correlation, indicating increased uterine artery resistance in women with previous cesarean sections [7,8], even if the differences were not statistically significant [9]. Conversely, other studies have found no significant differences or negative correlations [10,11].

Several theories have been proposed to explain this association. First, it is hypothesized that the uterine scar tissue resulting from a Cesarean section may lead to impaired vascular remodeling. This process is crucial during early pregnancy to ensure adequate blood flow to the placenta [12,13]. The scar may cause fibrosis and localized ischemia, reducing the elasticity of the uterine wall and increasing vascular resistance [14]. Another explanation proposed using animal studies is poor trophoblastic invasion, a process essential for establishing a healthy placental blood supply, which may be disrupted by cesarean-induced uterine scarring [15,16].

Based on the existing literature, we hypothesized that uterine artery resistance—particularly as measured by the Pulsatility Index (PI)—would be significantly higher in pregnant women with a history of cesarean section compared to those without. This hypothesis stems from studies suggesting impaired vascular remodellingand altered uteroplacental perfusion in scarred uteri. Clinically, identifying such associations could inform obstetric monitoring strategies, especially in high-risk pregnancies.

## Methods

### PECO for primary research question

We formulated the study question using the PECO framework [17] (Population, Exposure, Comparator, Outcome) as follows: Is a history of cesarean section associated with increased uterine artery resistance—measured by PI or RI—in subsequent pregnancies compared to women without such a history? Where Population (P): Pregnant women undergoing second-trimester Doppler ultrasound assessment; Exposure (E): History of previous cesarean section; Comparator (C): Women with no prior cesarean section; Outcome (O): Uterine artery resistance indices, including Pulsatility Index (PI) and Resistance Index (RI), assessed using color Doppler ultrasonography.

### Eligibility criteria

This systematic review was conducted according to Preferred Reporting Items for Systematic Reviews and Meta-Analyses (PRISMA) standards [18] and registered in Prospective Register of Systematic Reviews (PROSPERO) (CRD42024524343). We included all peer-reviewed original observational and interventional studies conducted in humans, as well as relevant academic theses (both published and unpublished) and full-text conference papers, provided they reported on uterine artery resistance measured by color Doppler ultrasonography in healthy pregnant women with and without a history of cesarean section. Eligible studies could be in any language, provided sufficient methodological and outcome data were accessible.

Studies were excluded if they lacked a comparison between women with and without prior cesarean section, did not report uterine artery resistance indices (PI or RI), provided insufficient data for analysis, or were non-original publications such as reviews, case reports, or abstracts without full texts.

### Information sources and search strategy

The databases PubMed, Scopus, Web of Science, and Embase were systematically searched for studies up to 6th April 2024. The authors concurred that Cesarean, Caesarean, "abdominal delivery", Doppler, "Uterine artery" and UtA be in the search query. Literature search was conducted in the title and abstract by mentioned search terms. The search strategy was tailored for each database to capture the most relevant studies (See S1 Table).

### Study selection

Two independent authors (Az.M and B.N) reviewed the title, abstract, and full text of retrieved studies after identifying and deleting duplicates by EndNote software version X8(Clarivate Analytics, Philadelphia, PA, USA). A third researcher (Ar.M) was consulted to resolve the conflict. Studies were included in the analysis if they reported Color Doppler ultrasonography results for uterine artery resistance in both patients with and without a history of cesarean section, either in the abstract or full text. To access non-English or unavailable studies, we email authors and journals [19–24].

### Risk of bias (ROB) and certainty assessment

All selected articles were assessed for the ROB using the Newcastle-Ottawa (NOS) checklist for observational studies [25]. Study quality was categorized into Good quality, Fair quality, and Poor quality based on the NOS checklist instruction. This tool evaluates three domains: A: Selection (up to 4 stars): Representativeness of the exposed cohort, selection of the non-exposed cohort, ascertainment of exposure, and confirmation that the outcome was not present at study start. B: Comparability (up to 2 stars): Assessment of whether the study controlled for confounding variables. C: Outcome (up to 3 stars): Assessment of outcome, adequacy of follow-up, and length of follow-up sufficient for outcomes to occur.

Two reviewers (Ar.M and Az.M) independently assigned stars to each item using a standardized NOS form. Disagreements were resolved through discussion or consultation with a third reviewer (Sh.Ch). Based on the total score out of 9

stars, studies were categorized as:Good quality (7–9 stars), Fair quality (4–6 stars), Poor quality (0–3 stars). All included studies scored 7 or higher and were rated as "Good" quality.

The certainty of evidence for each outcome was assessed using the GRADE (Grading of Recommendations, Assessment, Development and Evaluation) approach [26] b (Ar M). He performed both a manual GRADE assessment based on GRADE guidelines and a software-supported evaluation using GRADEpro GDT (https://gradepro.org). The manual assessment was used to provide contextual judgments for domains such as imprecision and inconsistency, particularly in interpreting clinical relevance and methodological consistency across studies.

## Summary measures

The PI and RI (Resistance Index) were considered the study outcomes and extracted from the papers. PI is a Doppler ultrasound measurement that reflects the pulsatility of blood flow, calculated as: PI = (Peak Systolic Velocity − End Diastolic Velocity)/ Mean Velocity. RI, also known as the Pourcelot index, is calculated as: RI = (Peak Systolic Velocity − End Diastolic Velocity)/ Peak Systolic Velocity [27]

## Data extraction

The author's name, publication years, age, Body Mass Index (BMI), Doppler week, PI, and RI were other variables extracted from the eligible studies and imported into the Excel database by two authors (B.N and E.S) independently and the discrepancy was dissolved by (Ar.M).

## Statistical method

Meta-analysis was conducted using STATA version 17 (StataCorp LLC, College Station, TX, USA) to synthesize data from the included studies. Each study's primary outcomes, including the PI and Resistance Index (RI), were extracted. We employed a random-effects model, specifically the DerSimonian-Laird method, to account for potential heterogeneity among studies. Hedges's g was calculated to measure the effect size, which estimates the standardized mean difference between the groups (women with and without a history of cesarean section).

Heterogeneity among the studies was assessed using the I² statistic, which describes the percentage of total variation across studies that is due to heterogeneity rather than chance. An I² value greater than 50% indicates substantial heterogeneity. Additionally, Cochran's Q test was used to test for heterogeneity. Potential publication bias was assessed using Egger's regression test. Given the small number of included studies, we supplemented Egger's regression test with visual inspection of funnel plots for each meta-analysis outcome, in accordance with PRISMA recommendations. A non-significant p-value (p > 0.05) in Egger's test suggests that there is no significant publication bias.

## Meta-regression and additional analyses

Meta-regression was performed to explore the influence of potential confounders on the observed effect sizes. This included variables such as gestational age at the time of Doppler assessment, maternal age, and BMI. Sensitivity analyses were also conducted by excluding studies that are the source of the heterogeneity to assess the robustness of the findings.

## Mean and standard deviation calculation using MoM

Since some studies reported the mean and standard deviation for the multiples of the median (MoM) of the PI, we generated simulated datasets reflecting the MoM distribution adjusted for gestational age using Gomez et al.'s reference values to estimate the mean and standard deviation of the PI values [28]. This Bayesian simulation process involved adjusting the MoM values for gestational age through regression analysis and summarizing the simulated PI values. The results from 2,000 simulations provided robust estimates for the mean and standard deviation of the PI, adjusted for gestational age.

## Results

A total of 958 records were identified through database searches, and one additional record was identified through other sources. After removing duplicates, 441 records were screened by title and abstract. Of these, 67 full-text articles were assessed for eligibility, and 61 were excluded with reasons. Ultimately, six studies met the inclusion criteria and were included in the final analysis. One of the included studies was conducted by our research team. Although we were aware of its findings during the review process, we followed a rigorous, predefined eligibility protocol and objective data extraction procedures to minimize bias.

The six included studies were published between 2011 and 2023 and originated from Turkey (n = 3), Iran (n = 1), Thailand (n = 1), and Norway (n = 1). All studies employed observational designs—either retrospective or prospective cohort studies. Collectively, they encompassed a total of 1,656 pregnant women, with individual study sample sizes ranging from 64 to 503. Uterine artery resistance was assessed using Doppler ultrasound, and outcomes were reported as either PI, RI, or both. All studies used standard gestational windows (11–32 weeks) for Doppler measurements. (Fig 1, Table 1). The PRISMA flow-chart illustrates the stringent selection process employed to ensure the inclusion of high-quality studies in our analysis. All six studies included in the meta-analysis were assessed for quality using an NOS checklist and were rated as "Good" based on their selection, comparability, and outcome criteria. (Table 2). The certainty of evidence for each outcome was assessed using the GRADE approach, both manually and through the GRADEpro GDT software (https://gradepro.org). For the association between prior cesarean section and increased uterine artery pulsatility index (PI), GRADEpro initially rated the certainty as low, consistent with default ratings for observational studies (S2 Table). However, based on consistent effect direction across studies, low heterogeneity ($I^2 = 26.6\%$), large pooled sample size (n = 1,656), and low risk of bias, we manually upgraded the certainty to moderate. Although the pooled effect estimate (Hedges's g = 0.15; 95% CI: 0.03 to 0.26) excluded the null, its small magnitude and use of log-transformed MoM values warranted cautious interpretation (Table 3). In contrast, the certainty of evidence for the resistance index (RI) was rated as very low in both manual and GRADEpro assessments, due to reliance on only two studies (n = 656), wide and non-significant confidence intervals (Hedges's g = 0.19; 95% CI: –0.06 to 0.43), possible inconsistency, and an inability to assess publication bias.

### Results for pulsatility index (PI) meta-analysis

The meta-analysis included six studies examining the PI in 1656 pregnant women with and without a history of cesarean section.

Using a random-effects model (DerSimonian–Laird method), the pooled effect size showed a statistically significant increase in PI for women with a history of cesarean section (Hedges's g = 0.15, 95% CI [0.03, 0.26], p = 0.01). The analysis revealed low heterogeneity among the studies ($I^2 = 26.60\%$, p = 0.23), indicating that the variability in effect sizes was primarily due to sampling error rather than true differences between the studies. The test for overall effect was significant (z = 2.45, p = 0.01), suggesting that the history of cesarean section is associated with an increased PI (Fig 2).

Assessment for publication bias using Egger's regression test indicated no significant small-study effects (p = 0.81), supporting the robustness of the results. The funnel plot did not show significant asymmetry, consistent with Egger's test results, further confirming the absence of substantial publication bias in the main analysis (Fig 3) and additional sensitivity analysis (S1 Fig). The studies were evenly distributed around the mean effect size, reinforcing the reliability of the meta-analysis findings.

Meta-regression was performed to explore the influence of potential confounders on the observed effect sizes, specifically considering age, BMI, and weeks of gestation. The meta-regression results indicated None of these factors showed a statistically significant impact on the effect size (p > 0.05), indicating that the association between cesarean section history and increased PI was not significantly influenced by these confounders. The residual heterogeneity was significant (Q_res = chi2(5) = 113.96, p < 0.0001), suggesting that other unmeasured factors may contribute to the variability in PI outcomes (S3 Table).

## PRISMA Flow Diagram

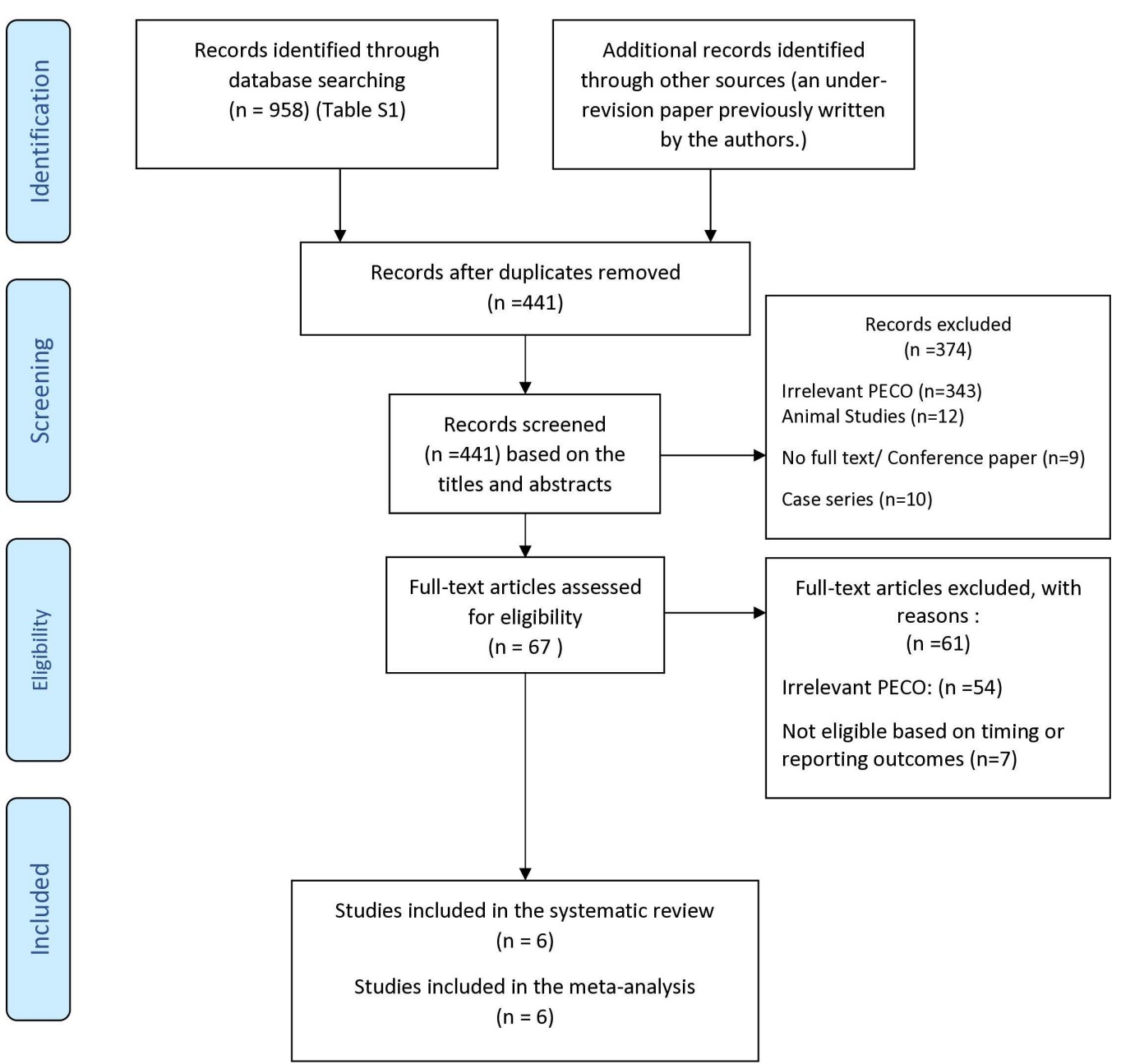

**Fig 1. PRISMA flow diagram of the systematic review.**

**Table 1. Characteristics of the studies which was included in the analysis.**

| Author's name | Year | Sample Size (Treatment/ Control) | Country | Eligibility Criteria | Exclusion Criteria | Gestation age (weeks)* | Study Design | Assessment Methodology | Key Findings |
|---|---|---|---|---|---|---|---|---|---|
| Flo et al [11] | 2014 | 32/32 | Norway | Pregnant women with and without Cesarean section history, singletons, 18–35 years old | Multiple pregnancies, chronic hypertension, diabetes | 22+0- 23+6 | Prospective Cohort | Color Doppler Ultrasonography | No significant difference in PI between groups |
| Torabi et al. [7] | 2019 | 200/200 | Iran | Pregnant women, singletons, at least one previous Cesarean section, 20–40 years old | Multiple pregnancies, pre-existing cardiovascular conditions | 18-22 | Prospective Cohort | Color Doppler Ultrasonography | Increased PI in Cesarean section group |
| Isikalan et al. [8] | 2020 | 80/73 | Turkey | Pregnant women, singletons, history of 1 or more Cesarean sections, no pregnancy complications | Multiple pregnancies, history of pre-eclampsia | 18-24 | Retrospective Cohort | Color Doppler Ultrasonography | Higher PI in Cesarean section group |
| Yapan et al. [9] | 2022 | 246/257 | Thailand | Pregnant women, singletons, history of Cesarean section or vaginal delivery, age 18–40 | Multiple pregnancies, pre-existing renal disease | 19–24 +6 | Prospective Cohort | Color Doppler Ultrasonography | Slightly increased PI in Cesarean section group |
| Aygun et al. [10] | 2023 | 126/102 | Turkey | Pregnant women, singletons, prior Cesarean section or vaginal delivery, no underlying health issues | Multiple pregnancies, thyroid disorders | 18-23 | Retrospective Cohort | Color Doppler Ultrasonography | No significant difference in PI between groups |
| Hashemi et al. [32] | 2024 | 154/154 | Iran | Pregnant women, singletons, history of Cesarean section, no chronic diseases | Multiple pregnancies, autoimmune diseases | 18-20 | Prospective Cohort | Color Doppler Ultrasonography | Increased PI in Cesarean section group |

*Gestational age was presented as Weeks (Day).

**Table 2. Assessment of the quality and risk of biases of the selected articles for the meta-analysis.**

| Study | Selection | | | | Comparability | | Outcome | | | | Total Score | Total Quality |
|---|---|---|---|---|---|---|---|---|---|---|---|---|
| | 1 | 2 | 3 | 4 | 1a | 1b | 1a | 1b | 2 | 3 | | |
| Flo et al. [11] | * | * | * | N/A | * | – | – | * | * | * | 7/8 | Good |
| Torabi et al. [7] | * | * | * | N/A | – | * | – | * | * | * | 7/8 | Good |
| Isikalan et al. [8] | * | * | * | N/A | – | * | – | * | * | * | 7/8 | Good |
| Yapan et al. [9] | * | * | * | N/A | * | – | – | * | * | * | 7/8 | Good |
| Aygun et al. [10] | * | * | * | N/A | * | – | – | * | * | * | 7/8 | Good |
| Hashemi et al. [32] | * | * | * | N/A | – | * | – | * | * | * | 7/8 | Good |

Studies scoring 7–9 stars were rated as good quality, 4–6 as fair quality, and 0–3 as poor quality.

**Table 3. The study certainly for two outcome variables of PI and RI using GRADE checklist.**

| Variable | GRADE Domains | Judgment | Notes |
|---|---|---|---|
| PI | Risk of Bias | Not serious | All included studies rated as "Good" quality (NOS); standardized Doppler protocols used. |
| | Inconsistency | Not serious | Low statistical heterogeneity (I²=26.6%) and consistent effect direction across studies. |
| | Indirectness | Not serious | The outcome (PI) directly addresses the PECO question; log-MoM transformation preserved relevance. |
| | Imprecision | Caution warranted | Although the confidence interval is statistically narrow and excludes the null, the small effect size (Hedges's g=0.15) and use of log-transformed MoM values may limit clinical interpretability. |
| | Publication Bias | Not serious | Egger's test (p=0.81) and funnel plot showed no evidence of publication bias. |
| RI | Risk of Bias | Not serious | Both studies scored well on NOS and used standard Doppler methodology. |
| | Inconsistency | Serious | Only two studies; overlapping CIs; true variability in effect cannot be confirmed. |
| | Indirectness | Not serious | Outcome directly reflects research question; no population or exposure mismatch. |
| | Imprecision | Serious | CI crosses null; effect size small and number of events limited. |
| | Publication Bias | Likely undetectable | Funnel plot and Egger's test not possible; bias cannot be ruled out with confidence. |

PI: Pulsatility Index, RI: Resistance Index.

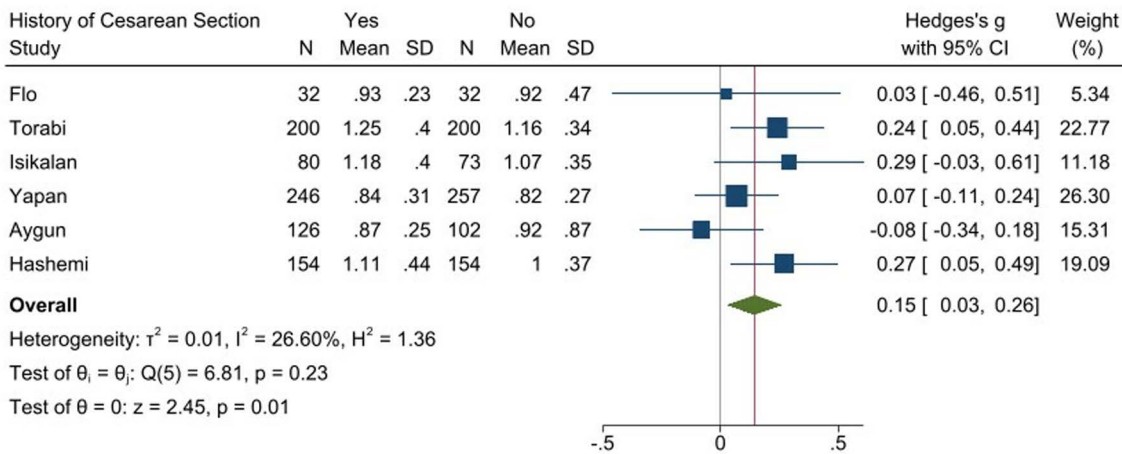

**Fig 2. Forest plot of Pulsatility Index (PI) meta-analysis.**

The sensitivity analysis for the PI indicates a statistically significant increase in PI for women with a history of cesarean section (Hedges's g=0.18, 95% CI [0.08, 0.29], p=0.00), with no observed heterogeneity among the studies (I²=0.00%) (S1 Fig).

## Resistance index (RI) meta-analysis

The meta-analysis of the RI included two studies, with a total sample size of 326 participants in the treatment group (history of cesarean section) and 330 participants in the control group (no history of cesarean section). The pooled effect

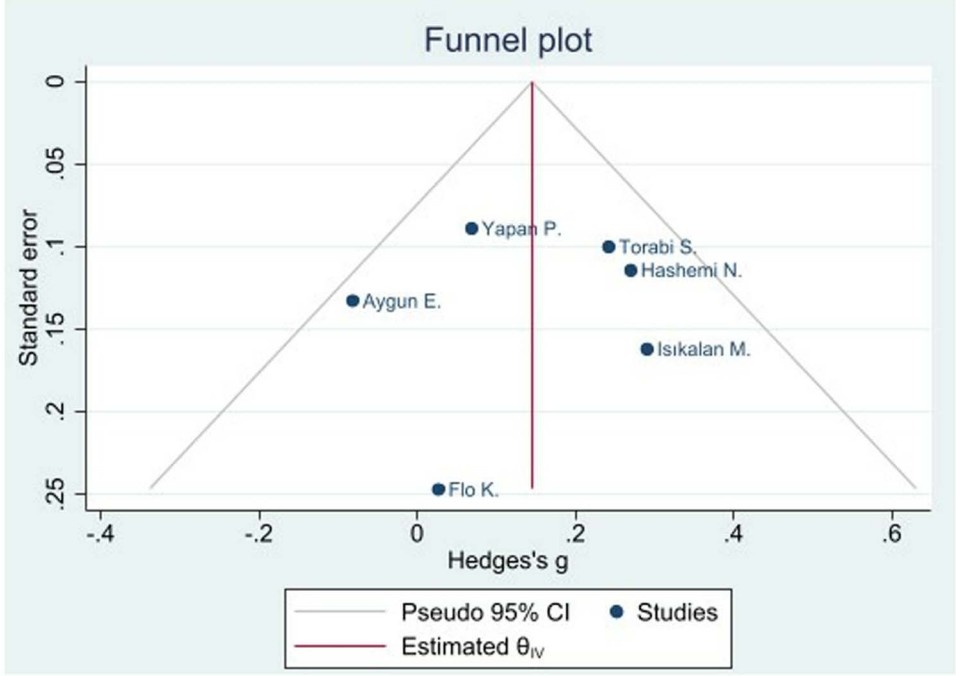

**Fig 3. Funnel plot of publication bias for Pulsatility Index (PI).**

size, calculated using a random-effects model, showed a small increase in RI for women with a history of cesarean section (Hedges's g=0.19, 95% CI [−0.06, 0.43], p=0.13). The heterogeneity among the studies was moderate (I²=48.08%, p=0.17), indicating some variability in the results across studies. The test of overall effect (z=1.50, p=0.13) suggests that the increase in RI was not statistically significant (Fig 4). For the RI outcome, no definitive interpretation could be made regarding publication bias due to the small number of studies (n=2).

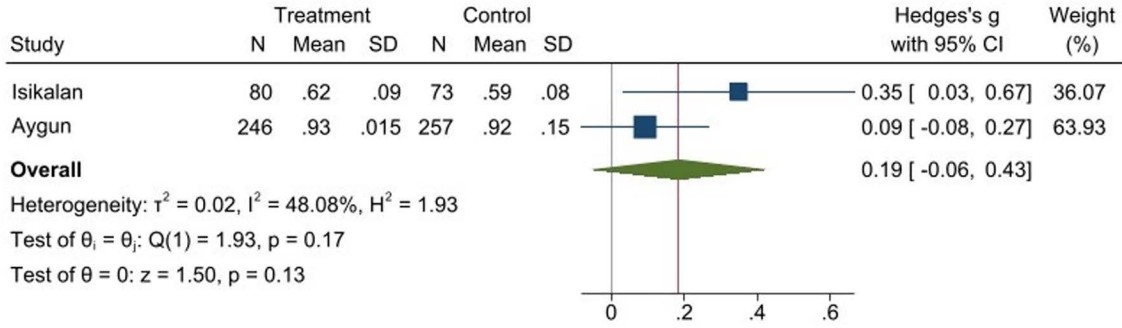

**Fig 4. Forest plot of Resistance Index (RI) meta-analysis.**

## Discussion

### Summary of main results

The results of our meta-analysis indicate a statistically significant increase in the PI in pregnant women with a history of cesarean section compared to those without. This finding is consistent across the studies included in the analysis. However, we needed more data to comprehensively analyse the RI.

### General interpretation of the results in the context of other evidence

Our findings align with previous studies that have evaluated the impact of prior cesarean sections on maternal and perinatal outcomes in subsequent pregnancies. For instance, a review by Silver et al. reported that previous cesarean deliveries were associated with increased risks of placental complications, uterine rupture, and adverse neonatal outcomes in subsequent pregnancies [29]. Additionally, Lumbiganon et al. found that women with a history of cesarean section had higher rates of perinatal complications such as low birth weight and neonatal intensive care unit admissions [30]. These findings corroborate our meta-analysis results, which indicate that previous cesarean sections are associated with increased uterine artery resistance, potentially leading to higher rates of adverse outcomes.

The potential mechanism behind the increased PI and its association with adverse outcomes such as preeclampsia, intrauterine growth restriction (IUGR), and preterm labor could be attributed to the impaired trophoblastic invasion of spiral arteries. This results in inadequate remodellingof these arteries, leading to higher resistance and reduced uteroplacental blood flow [31]. The studies included in our meta-analysis reported various adverse perinatal outcomes linked to high PI. As Hashemi et al. found, increased PI correlated with higher incidences of preeclampsia and IUGR, highlighting the clinical significance of monitoring PI in pregnancies with a history of cesarean section [32].

Our study primarily focused on the second trimester, contrasting with studies like those by Filho et al. (2011) and Zarean and Shabaninia (2018), which evaluated uterine artery resistance in the third trimester. Filho et al. found no significant changes in the uterine artery PI between women with elective cesarean sections and those with cesarean sections performed during labor. They concluded that the uterine matrix circulation is robust enough to withstand the scarring from cesarean sections without significant alterations detectable by Doppler velocimetry [33]. On the other hand, Zarean and Shabaninia observed that high uterine artery PI at 30–34 weeks gestation was associated with adverse perinatal outcomes, including small-for-gestational-age fetuses and preeclampsia, although its predictive power alone was limited [34].

Overall, our findings support the need for heightened surveillance of uterine artery resistance in pregnant women with a history of cesarean section, particularly focusing on PI as a critical marker. Future research should include comprehensive data on both PI and RI, and explore the mechanisms further to improve predictive models for adverse perinatal outcomes.

The evaluation of certainty highlights important contrasts between the two outcomes assessed. For PI, while the GRADEpro software initially rated the certainty of evidence as low—due to the observational nature of the included studies—a manual reassessment supported an upgrade to moderate certainty, based on consistent direction of effect, high methodological quality, and low heterogeneity ($I^2 = 26.6\%$). Although the observed effect size was small and based on log-transformed multiples of the median (MoM), its statistical significance and precision increase confidence in the validity of the association. These findings suggest a likely, though modest, physiological link between prior cesarean section and increased uterine artery resistance. In contrast, the very low certainty for RI, as confirmed by both manual and GRADEpro assessments, reflects the limited number of studies, imprecise and non-significant pooled estimates, and the inability to evaluate publication bias. This disparity reinforces the need for additional high-quality observational research to clarify whether RI can serve as a valid surrogate marker of uteroplacental resistance in this population.

## Potential biases and limitations

One of the primary limitations of our study is the lack of a sufficient number of relevant studies. The limited availability of studies reporting the RI prevented us from providing more comprehensive evidence regarding this metric. Additionally, many studies reported the PI in terms of multiples of the median (MoM) instead of providing the actual mean and standard deviation. This necessitated additional calculations and assumptions that could introduce bias and affect the accuracy of our meta-analysis. One of the studies included in the meta-analysis was conducted by our research team. While we were aware of its results during the review process, we adhered strictly to predefined eligibility criteria and objective data extraction procedures to minimize bias and ensure methodological rigor. Another significant limitation is the failure of some studies to disclose the exact number of previous cesarean sections, which is a crucial variable for assessing the impact on uterine artery resistance. During the review process, we encountered difficulties in accessing the full texts of some papers that initially appeared relevant. Despite our efforts to request these articles from journals and authors, we could not obtain the full text. This may have led to the exclusion of potentially relevant data, impacting the comprehensiveness of our review. A further limitation of this review is the geographical concentration of the included studies, with five of the six originating from Asian countries. This may limit the generalizability of the findings to other populations, particularly those in regions with different obstetric practices including rate of cesarean section and Doppler ultrasound cost and availability. For instance, countries such as Turkey, Iran, and China have among the highest cesarean section rates globally, with a substantial proportion being elective rather than medically indicated procedures [35,36]. In parallel, uterine artery Doppler ultrasound is more routinely used in parts of Asia for second-trimester screening [37], particularly for the early identification of preeclampsia and placental insufficiency, whereas its use in many Western countries is typically limited to high-risk pregnancies [38,39]. These regional differences in both clinical practice and patient populations may influence the observed associations and should be considered when interpreting the applicability of our findings in other healthcare settings. To the best of our knowledge, this is the first systematic review and meta-analysis to comprehensively evaluate the impact of prior cesarean delivery on uterine artery Doppler indices, addressing a critical gap in the obstetric literature; additionally, the standardized log-transformation of Doppler values into multiples of the median (MoM) across studies enhanced the comparability and precision of the pooled estimates.

## Implications for practice and future research

The results of our meta-analysis suggest an association between a history of cesarean section and increased uterine artery resistance. This finding highlights the importance of careful monitoring of uterine artery resistance in pregnant women with a history of cesarean section to mitigate potential adverse outcomes. Clinicians should consider this factor when managing pregnancies following a cesarean section.

Future research should focus on including larger sample sizes and diverse populations to validate our findings, using standardized Doppler protocols across regions. Studies should also ensure the disclosure of key clinical variables, such as the number and indication of previous cesarean sections, to allow for more precise subgroup analyses. Additionally, facilitating open access to full-text data would improve the completeness of evidence synthesis. Importantly, future studies should investigate the clinical implications of elevated uterine artery resistance in women with prior cesarean deliveries—particularly regarding outcomes such as preeclampsia, fetal growth restriction, and placental insufficiency—to better guide prenatal care.

## Conclusion

This meta-analysis suggests that women with a history of cesarean section may exhibit slightly increased uterine artery resistance, as measured by Doppler indices. While the observed association is modest, it may have implications for uteroplacental perfusion and pregnancy outcomes. These findings highlight the potential value of closer monitoring in

pregnancies following cesarean delivery, although further research is needed to clarify the clinical significance and guide management strategies.

## Supporting information

**S1 Table. Detail of search strategies in each database and corresponding queries.**
(DOCX)

**S2 Table. The evaluation of certainty using the GradePro software and also manually.**
(DOCX)

**S3 Table. Results of meta-regression analysis to identify potential confounding effects.**
(DOCX)

**S1 Fig. Sensitivity analysis.** A: Forest plot of the PI meta-analysis following removal of the source of the heterogeneity. B: Funnel plot of the PI meta-analysis following removal of the source of the heterogeneity.
(PDF)

## Author contributions

**Conceptualization:** Arash Mohazzab, Shahla Chaichian.

**Formal analysis:** Arash Mohazzab, Azar Mohammadzadeh, Safdar Masoumi, Elham Shoraka.

**Investigation:** Azar Mohammadzadeh, Banafsheh Nikfar, Safdar Masoumi, Elham Shoraka.

**Methodology:** Arash Mohazzab, Shahla Chaichian.

**Supervision:** Shahla Chaichian.

**Writing – original draft:** Banafsheh Nikfar, Neda Hashemi.

**Writing – review & editing:** Neda Hashemi.

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
