## [Decision Letter · Decision Letter 0]

Dear Dr. Chaichian,

Thank you for submitting your manuscript to PLOS ONE. After careful consideration, we feel that it has merit but does not fully meet PLOS ONE’s publication criteria as it currently stands. Therefore, we invite you to submit a revised version of the manuscript that addresses the points raised during the review process.

We look forward to receiving your revised manuscript.

Kind regards,

Mohammad Haddadi, MD

Academic Editor

PLOS ONE

Journal Requirements:

4. Please include captions for your Supporting Information files at the end of your manuscript, and update any in-text citations to match accordingly. Please see our Supporting Information guidelines for more information: http://journals.plos.org/plosone/s/supporting-information .

5. As required by our policy on Data Availability, please ensure your manuscript or supplementary information includes the following:

Additional Editor Comments (if provided):

Your manuscript and topic are suitable. However, I believe several modifications are necessary for improvement:

1. Avoid abbreviations in the abstract, such as “PI,” unless they are first fully spelled out.

2.Review the entire manuscript for proper capitalization and make corrections where needed.

3. Rewrite the conclusion section in the abstract—it is currently unclear and needs to be more concise and informative.

4. Do not separate the introduction into “Rationale” and “Objective” sections. It is more standard to present the Introduction as a single, cohesive section.

5. Avoid using abbreviations without first providing their full form, such as “C-section” (line 51), “BMI” (line 86), and “ROB” (line 82).

6. Expand the Introduction by discussing other tools and assessments. Explain the significance of comparing uterine artery resistance with C-section history and clarify why this comparison matters. The importance of the study should be well articulated.

7. Include citations for reporting guidelines such as PRISMA. Do not use abbreviations like PRISMA before introducing and citing them properly.

8. Describe the Risk of Bias (ROB) assessment in detail in both the Methods and Results sections. Explain how the assessment was conducted and highlight specific limitations in the included studies. Do not just report qualitative ratings—also provide final scores alongside terms like “good,” “fair,” etc.

9. Specify the software package used for analysis in the Methods section.

10. As previously mentioned, discuss the Risk of Bias in detail in the Results section, including study-specific assessments.

11. Avoid repeating abbreviations like “Pulsatility Index (PI)” once already defined (e.g., in lines 129, 132, etc.).

12. Clearly state the aim of the study at the beginning of the Discussion, linking it directly to the outcomes.

13. Include full descriptions of any abbreviations used in tables in the footnotes.

14. Describe the Doppler assessment methodology for each included study—how it was performed and according to which guidelines.

17. According to the AMSTAR guidelines, it is recommended to report the source of funding for each included study, as this adds transparency.

Reviewers' comments:

Reviewer's Responses to Questions

**Comments to the Author**

1. Is the manuscript technically sound, and do the data support the conclusions?

Reviewer #1: Yes

Reviewer #2: Yes

2. Has the statistical analysis been performed appropriately and rigorously?

Reviewer #1: Yes

Reviewer #2: No

3. Have the authors made all data underlying the findings in their manuscript fully available?

Reviewer #1: Yes

Reviewer #2: Yes

4. Is the manuscript presented in an intelligible fashion and written in standard English?

Reviewer #1: Yes

Reviewer #2: Yes

Reviewer #1: Overall Appraisal

This meta-analysis and systematic review addresses a relevant clinical question by examining the association between prior cesarean section and uterine artery resistance in subsequent pregnancies. The study is well-structured, with a clear methodology, including a comprehensive literature search, independent study selection, and appropriate risk-of-bias assessment. However, before considering it for publication, the following key modifications are required:

Comments to The Authors

Abstract

• Line 38-40: It is not clear whether the authors are referring to PI or RI as a measure of uterine artery resistance here. I strongly recommend on reporting the results from the meta-analysis for both measures here.

Introduction

• Line 49-51: Please provide robust references for each cited sentence.

• Line 51: Please ensure consistent use of "cesarean section" throughout the manuscript, as "C-section" appears here.

• Line 65: Please state your hypothesis regarding the meta-analysis results based on existing literature, and discuss the potential clinical and research implications of your findings.

Methods

• Line 67: Before defining the eligibility criteria, please define your research question following the PICO framework.

• Line 68: Please define the abbreviations PRISMA and PROSPERO.

• Line 67-70: The current eligibility criteria lack sufficient precision for proper evaluation and require clarification. Please explicitly specify:

- The types of studies to be included (e.g., study designs, publication types)

- Detailed exclusion criteria (e.g., handling of non-peer-reviewed literature, non-original articles, non-human studies)

- Clear inclusion/exclusion parameters for different study designs

• Line 78: Please mention the EndNote version and company name.

• The authors must provide a new subsection describing the process of data extraction from each included study and mention what kind of data were systematically extracted from each study. Also, the review outcomes (RI and PR) should be defined here using robust references, e.g., ACOG or UOG guidelines.

• Line 82: Please define the abbreviation ROB.

• Line 82-83: The authors must define how each item of NOS was answered and how the overall appraisal of the quality of each study was made.

• Line 96: While the authors' use of Egger's test for publication bias assessment is appropriate, its reliability may be limited given the small number of included studies (n=6). We strongly recommend supplementing these results with funnel plots for each meta-analysis, as they would provide (1) visual objectivity for assessing symmetry and potential bias, (2) complementary evidence to address statistical limitations with small study counts, and (3) greater transparency to enhance reproducibility of conclusions, thereby strengthening the overall bias assessment in line with PRISMA guidelines.

• Line 99: Please define the abbreviation BMI.

• Line 100: Please clearly mention how the results of the meta-regression was interpreted and what measures were calculated for this purpose.

• Line 100: Please provide funnel plots for the leave-one-out sensitivity analyses.

• Line 103: Please provide a citation for the mentioned reference.

Results

• Line 113: Before reporting the meta-analysis results, it is necessary to systematically review the characteristics of the included studies, encompassing the overall included sample size, the sample size ranges, publication dates, countries of origin, study designs, measured outcomes, etc.

• Line 112-138: Please clearly mention how many studies and how many pregnancies were included in each meta-analysis.

Discussion

• Line 144: The authors only compared their findings with prior studies, while no interpretation was provided on how a history of CS can be associated with elevated uterine artery resistance in the current pregnancy. It is vital to discuss the potential pathophysiology behind this phenomenon.

• The discussion section appears underdeveloped, as evidenced by the limited number of references for a systematic review. This scarcity suggests insufficient engagement with the existing literature to adequately contextualize and interpret the findings. To strengthen the manuscript, the discussion should be expanded to: (1) incorporate more relevant studies to support key interpretations, (2) compare findings with previous meta-analyses on similar topics, and (3) address contradictory evidence in the literature. A more comprehensive discussion with additional references would enhance the scholarly rigor and provide readers with a better perspective on how these results advance current understanding.

Reviewer #2: General Comments:

The authors have done a commendable job in compiling and presenting the data. However, the manuscript would benefit from a thorough revision for grammar, sentence structure, and overall clarity to enhance readability and academic tone.

Introduction:

• Lines 50–51: Please provide a citation to support the statement.

• Line 57: Since multiple studies are referenced and no significant or negative outcomes are stated, more than one citation is needed to substantiate the claim.

Methods:

• Please define the abbreviations PRISMA and PROSPERO upon first use.

• Indicate the date on which the literature search was conducted.

• Line 78: Specify the version of the software which was used.

• When listing author contributions, include their initials in parentheses, e.g., (A.B. and C.D.).

• In the “Summary of Measures” section, clarify whether data extraction was performed by one or two researchers. Additionally, explain how discrepancies between reviewers were resolved.

• Since the results section includes a funnel plot, its use should also be described in the methods section.

Results:

• Clearly state the number of articles initially retrieved, how many were screened by title and abstract, how many underwent full-text review, and how many were ultimately included in the analysis.

• In the PRISMA flow diagram, one study is listed as coming from another source. Please clarify the origin of this study.

Discussion:

• Line 157: A citation is required to support the statement.

• The quality of evidence is typically evaluated using the GRADE tool. As your manuscript claims adherence to PRISMA guidelines, inclusion of GRADE assessment is recommended.

• Under limitations, it would be appropriate to mention the limited geographical generalizability of the findings, given that five studies were from Asia and one from Europe.

• Consider including the strengths of your study alongside the limitations.

• It would be beneficial to include recommendations for future research directions in the conclusion section.

Table 1:

• In Row 2, the reference appears in the middle of the author’s name. Please correct this.

Figure 1:

• Indicate how many results were retrieved from each database.

• Specify the reasons for exclusion and the number of studies excluded for each reason.

**Do you want your identity to be public for this peer review?** For information about this choice, including consent withdrawal, please see our Privacy Policy

Reviewer #1: **Yes: ** Mohammadamin Parsaei

Reviewer #2: No

---

## [Author Response · Author response to Decision Letter 1]

25 Apr 2025

We would like to convey our special thanks to the editor and all the reviewers for their time and efforts to read this manuscript and the valuable comments they have made to improve this paper.

Hereby, the point to point answers and correction are listed below and are sent back for consideration. All changes were applied and highlighted in the manuscript.

General Comments:

The authors have done a commendable job in compiling and presenting the data. However, the manuscript would benefit from a thorough revision for grammar, sentence structure, and overall clarity to enhance readability and academic tone.

Introduction:

• Lines 50–51: Please provide a citation to support the statement.

The references were added. (Please see reference 1)

Line 57: Since multiple studies are referenced and no significant or negative outcomes are stated, more than one citation is needed to substantiate the claim.

The references were added. (Please see reference 11)

Methods:

• Please define the abbreviations PRISMA and PROSPERO upon first use.

The abbreviations were added. Please see the lines 77 and 78.

• Indicate the date on which the literature search was conducted.

It head been previously written in the text as April 2024 (Please see the line 83).

• Line 78: Specify the version of the software which was used.

It was added. Please see the line 89.

• When listing author contributions, include their initials in parentheses, e.g., (A.B. and C.D.).

The requested changes was applied. Please see the line 88

• In the “Summary of Measures” section, clarify whether data extraction was performed by one or two researchers. Additionally, explain how discrepancies between reviewers were resolved.

The requested changes was applied. Please see the line 117

• Since the results section includes a funnel plot, its use should also be described in the methods section.

The requested changes was applied. Please see the line 128

Results:

• Clearly state the number of articles initially retrieved, how many were screened by title and abstract, how many underwent full-text review, and how many were ultimately included in the analysis.

The requested changes was applied. Please see the line 142

• In the PRISMA flow diagram, one study is listed as coming from another source. Please clarify the origin of this study.

• The requested changes was applied. Please see the line 145

Discussion:

• Line 157: A citation is required to support the statement.

The statement was generated based on result of two abovementioned studies. Was sited again. Please see the line 202.

• The quality of evidence is typically evaluated using the GRADE tool. As your manuscript claims adherence to PRISMA guidelines, inclusion of GRADE assessment is recommended.

Since all included study are observational we used NOS checklists for the quality assessment which is specific with this study type. Nevertheless, based on useful comment of the reviewer we add the results of the GRADE assessment for the overall certainty of the results and added it to the method ( line 100), result( line 155) and discussion (line 220) sections beside table 3.

• Under limitations, it would be appropriate to mention the limited geographical generalizability of the findings, given that five studies were from Asia and one from Europe.

The limitation was mentioned and was discussed in detail (line 250).

•

• Consider including the strengths of your study alongside the limitations.

• The limitation was mentioned (line 260).

• It would be beneficial to include recommendations for future research directions in the conclusion section.

The strength was mentioned (line 259).

Table 1:

• In Row 2, the reference appears in the middle of the author’s name. Please correct this.

• Corrected

Figure 1:

• Indicate how many results were retrieved from each database.

• It was added to the table S1 (supplement)

• Specify the reasons for exclusion and the number of studies excluded for each reason.

• It was added to the figure 1

• Reviewer #1: Overall Appraisal

This meta-analysis and systematic review addresses a relevant clinical question by examining the association between prior cesarean section and uterine artery resistance in subsequent pregnancies. The study is well-structured, with a clear methodology, including a comprehensive literature search, independent study selection, and appropriate risk-of-bias assessment. However, before considering it for publication, the following key modifications are required:

Comments to The Authors

Abstract

• Line 38-40: It is not clear whether the authors are referring to PI or RI as a measure of uterine artery resistance here. I strongly recommend on reporting the results from the meta-analysis for both measures here.

The requested change was applied. Please see the line 38-41

•

Introduction

• Line 49-51: Please provide robust references for each cited sentence.

The requested change was applied. Please see ref 1.

• Line 51: Please ensure consistent use of "cesarean section" throughout the manuscript, as "C-section" appears here.

It was corrected.

• Line 65: Please state your hypothesis regarding the meta-analysis results based on existing literature, and discuss the potential clinical and research implications of your findings.

The requested text was added. Please see the line 66

Methods

• Line 67: Before defining the eligibility criteria, please define your research question following the PICO framework.

The requested change was applied. Please see the line 71

• Line 68: Please define the abbreviations PRISMA and PROSPERO.

The requested change was applied. Please see the line 77

• Line 67-70: The current eligibility criteria lack sufficient precision for proper evaluation and require clarification. Please explicitly specify:

- The types of studies to be included (e.g., study designs, publication types)

- Detailed exclusion criteria (e.g., handling of non-peer-reviewed literature, non-original articles, non-human studies)

- Clear inclusion/exclusion parameters for different study designs.

The requested change was applied. Please see the line 78-82.

• • Line 78: Please mention the EndNote version and company name.

The requested change was applied. Please see the line 89

• The authors must provide a new subsection describing the process of data extraction from each included study and mention what kind of data were systematically extracted from each study. Also, the review outcomes (RI and PR) should be defined here using robust references, e.g., ACOG or UOG guidelines.

The requested change was applied. Please see the line 108-114

• Line 82: Please define the abbreviation ROB.

The requested change was applied. Please see the line 93

• Line 82-83: The authors must define how each item of NOS was answered and how the overall appraisal of the quality of each study was made.

The requested change was applied. Please see the line 94-102

• Line 96: While the authors' use of Egger's test for publication bias assessment is appropriate, its reliability may be limited given the small number of included studies (n=6). We strongly recommend supplementing these results with funnel plots for each meta-analysis, as they would provide (1) visual objectivity for assessing symmetry and potential bias, (2) complementary evidence to address statistical limitations with small study counts, and (3) greater transparency to enhance reproducibility of conclusions, thereby strengthening the overall bias assessment in line with PRISMA guidelines.

The requested change was applied. Please see the line 126-128 and Figure S2

• Line 99: Please define the abbreviation BMI.

The requested change was applied. Please see the line 133

• Line 100: Please provide funnel plots for the leave-one-out sensitivity analyses

The requested change was applied. Please see the Figure S2.

• Line 103: Please provide a citation for the mentioned reference.

The requested change was applied. Please see the line 138

Results

• Line 113: Before reporting the meta-analysis results, it is necessary to systematically review the characteristics of the included studies, encompassing the overall included sample size, the sample size ranges, publication dates, countries of origin, study designs, measured outcomes, etc.

It was added to the result section. Please see the lines 142-146

• Line 112-138: Please clearly mention how many studies and how many pregnancies were included in each meta-analysis.

The requested change was applied. Please see the line 148.

Discussion

• Line 144: The authors only compared their findings with prior studies, while no interpretation was provided on how a history of CS can be associated with elevated uterine artery resistance in the current pregnancy. It is vital to discuss the potential pathophysiology behind this phenomenon.

It had been discussed in the line 201-204

• The discussion section appears underdeveloped, as evidenced by the limited number of references for a systematic review. This scarcity suggests insufficient engagement with the existing literature to adequately contextualize and interpret the findings. To strengthen the manuscript, the discussion should be expanded to: (1) incorporate more relevant studies to support key interpretations, (2) compare findings with previous meta-analyses on similar topics, and (3) address contradictory evidence in the literature. A more comprehensive discussion with additional references would enhance the scholarly rigor and provide readers with a better perspective on how these results advance current understanding.

Thank you for tour comment. We tried to find and discuss all relevant studies. The scarcity is directly related to the low number

• Reviewer #2: General Comments:

The authors have done a commendable job in compiling and presenting the data. However, the manuscript would benefit from a thorough revision for grammar, sentence structure, and overall clarity to enhance readability and academic tone.

Introduction:

• Lines 50–51: Please provide a citation to support the statement.

The reference was added. Please fine the ref (1)

• Line 57: Since multiple studies are referenced and no significant or negative outcomes are stated, more than one citation is needed to substantiate the claim.

Please see the ref number (11)

Methods:

• Please define the abbreviations PRISMA and PROSPERO upon first use.

The requested change was applied. Please see the line 77

• Indicate the date on which the literature search was conducted.

The date had been previously mentioned in the line 83.

• Line 78: Specify the version of the software which was used.

The requested change was applied. Please see the line 89

• When listing author contributions, include their initials in parentheses, e.g., (A.B. and C.D.).

The requested change was applied. Please see the line 88 and 103

• In the “Summary of Measures” section, clarify whether data extraction was performed by one or two researchers. Additionally, explain how discrepancies between reviewers were resolved.

The requested change was applied. Please see the line 117

• Since the results section includes a funnel plot, its use should also be described in the methods section.

The requested change was applied. Please see the line 127

Results:

• Clearly state the number of articles initially retrieved, how many were screened by title and abstract, how many underwent full-text review, and how many were ultimately included in the analysis.

The requested change was applied. Please see the line 133

• In the PRISMA flow diagram, one study is listed as coming from another source. Please clarify the origin of this study.

The requested change was applied. Please see the line 142.

Discussion:

• Line 157: A citation is required to support the statement.

The requested change was applied. Please see the line 206

• The quality of evidence is typically evaluated using the GRADE tool. As your manuscript claims adherence to PRISMA guidelines, inclusion of GRADE assessment is recommended.

Since all included study are observational we used NOS checklists for the quality assessment which is specific with this study type. Nevertheless, based on useful comment of the reviewer we add the results of the GRADE assessment for the overall certainty of the results and added it to the method ( line 100), result( line 155) and discussion (line 220) sections beside table 3.

• Under limitations, it would be appropriate to mention the limited geographical generalizability of the findings, given that five studies were from Asia and one from Europe.

The limitation was mentioned and was discussed in detail (line 250).

• Consider including the strengths of your study alongside the limitations.

The requested text was added. Please see the line 258

• It would be beneficial to include recommendations for future research directions in the conclusion section.

The requested text was added. Please see the line 268

Table 1:

• In Row 2, the reference appears in the middle of the author’s name. Please correct this.

it was corrected

Figure 1:

• Indicate how many results were retrieved from each database.

The requested change was applied.

• Specify the reasons for exclusion and the number of studies excluded for each reason.

• The requested change was applied.

---

## [Decision Letter · Decision Letter 1]

Thank you for submitting your manuscript to PLOS ONE. After careful consideration, we feel that it has merit but does not fully meet PLOS ONE’s publication criteria as it currently stands. Therefore, we invite you to submit a revised version of the manuscript that addresses the points raised during the review process.

We look forward to receiving your revised manuscript.

Kind regards,

Mohammad Haddadi, M.D.

Academic Editor

PLOS ONE

Journal Requirements:

Additional Editor Comments:

Dear Authors,

Thank you for your efforts. Please address the following revisions:

1. Remove subheadings in the Introduction section.

2. Revise the Background of the Abstract. It should state the objective, not the results. For example, instead of "We included studies that reported…", write "We aimed to evaluate…".

3. Review grammar and spelling throughout the manuscript. For instance, clarify whether "PECO" is correctly used.

4. Ensure consistent use of abbreviations. For example, BMI is introduced as an abbreviation in line 116, but the full form appears again in line 133. Please revise for consistency.

5. Use a consistent form of English (either British or American) throughout the manuscript. For example, "Localized" in line 61 is American English, while "Analyses" in line 131 is British. Please standardize the language throughout.

6. Do not repeat the full form after defining an abbreviation. For instance, in lines 156–157, once PI and RI are defined, only use the abbreviations thereafter.

7. Move the 'Quality of the Evidence' section to the Results. It should be interpreted in the Discussion, not presented there as a result.

8. In Table 2, include the final score for each study and provide an interpretation guide in a footnote.

9. PRISMA diagram: In the final cell, specify the number of studies included in the systematic review and the meta-analysis—even if the numbers are the same.

Reviewers' comments:

Reviewer's Responses to Questions

**Comments to the Author**

Reviewer #1: All comments have been addressed

Reviewer #2: (No Response)

2. Is the manuscript technically sound, and do the data support the conclusions?

Reviewer #1: Yes

Reviewer #2: Yes

3. Has the statistical analysis been performed appropriately and rigorously?

Reviewer #1: Yes

Reviewer #2: Yes

4. Have the authors made all data underlying the findings in their manuscript fully available?

Reviewer #1: Yes

Reviewer #2: Yes

5. Is the manuscript presented in an intelligible fashion and written in standard English?

Reviewer #1: Yes

Reviewer #2: Yes

Reviewer #1: The comments have addressed my comments in a acceptable fashion. However, I recommend on the following modifications, before considering it for publication:

- Line 37: remove " to assess PI".

- Line 42: Lower your tone as the current evidence is really limited and further research in warranted.

- Line 72: E in PECO is for Exposure not Intervention

- Line 76: The exclusion criteria is not defined yet!

- Line 77: Please cite the main article for PRISMA 2020 (Page et al. 2021)

- Please use GRADEPro to make a solid table for your evidence certainty assessments.

Reviewer #2: In Line 72, the manuscript mischaracterizes the PECO model as being based on interventions; however, PECO is more appropriately used for observational studies and is structured around exposure rather than intervention. This should be corrected, and an appropriate citation supporting the use of the PECO framework should be added.

The study question is presented using the PECO format, but the individual components—Population, Exposure, Comparator, and Outcome—are not clearly defined. Each element should be explicitly stated to ensure clarity and methodological transparency.

The manuscript refers to the PRISMA guideline but does not provide a proper citation. A formal reference to the most current version of the PRISMA guideline (e.g., PRISMA 2020) should be included.

The phrase “including papers, published and unpublished thesis” is imprecise. It should be revised to clearly describe the types of literature included in the review, specifying the nature of the unpublished materials and the sources used to retrieve them.

The response to the earlier comment regarding the timing of the literature search is insufficient. The manuscript must explicitly indicate the date on which the literature search was conducted by your team for each database or platform. Merely referencing Line 83 is inadequate; the information should be clearly presented in the methods section.

As noted previously, the flow diagram in Figure 1 lacks specificity regarding exclusion criteria. The reasons for exclusion should be clearly stated both at the title and abstract screening stage and at the full-text review stage, in line with best practices for systematic review reporting.

**Do you want your identity to be public for this peer review?** For information about this choice, including consent withdrawal, please see our Privacy Policy

Reviewer #1: No

Reviewer #2: No

---

## [Author Response · Author response to Decision Letter 2]

11 May 2025

Dear Authors,

Thank you for your efforts. Please address the following revisions:

1. Remove subheadings in the Introduction section.

The correction was applied.

2. Revise the Background of the Abstract. It should state the objective, not the results. For example, instead of "We included studies that reported…", write "We aimed to evaluate…".

The requested change was made.

3. Review grammar and spelling throughout the manuscript. For instance, clarify whether "PECO" is correctly used.

The requested change was made.

4. Ensure consistent use of abbreviations. For example, BMI is introduced as an abbreviation in line 116, but the full form appears again in line 133. Please revise for consistency.

The requested change was applied.

5. Use a consistent form of English (either British or American) throughout the manuscript. For example, "Localized" in line 61 is American English, while "Analyses" in line 131 is British. Please standardize the language throughout.

The requested change was applied.

6. Do not repeat the full form after defining an abbreviation. For instance, in lines 156–157, once PI and RI are defined, only use the abbreviations thereafter.

The requested change was applied.

7. Move the 'Quality of the Evidence' section to the Results. It should be interpreted in the Discussion, not presented there as a result.

We relocated the Quality of the Evidence to the results section but a paragraph was dedicated to discuss the results in the discussion section too.

8. In Table 2, include the final score for each study and provide an interpretation guide in a footnote.

The requested change was applied.

9. PRISMA diagram: In the final cell, specify the number of studies included in the systematic review and the meta-analysis—even if the numbers are the same.

The requested change was applied.

Review Comments to the Author

Reviewer #1: The comments have addressed my comments in a acceptable fashion. However, I recommend on the following modifications, before considering it for publication:

- Line 37: remove " to assess PI".

The requested correction was applied.

- Line 42: Lower your tone as the current evidence is really limited and further research in warranted.

We rewrited the conclusion

- Line 72: E in PECO is for Exposure not Intervention

Corrected.

- Line 76: The exclusion criteria is not defined yet!

Exclusion criteria were added.

- Line 77: Please cite the main article for PRISMA 2020 (Page et al. 2021)

The citation was added.

- Please use GRADEPro to make a solid table for your evidence certainty assessments.

We used GRADEpro for baseline assessment and manually adjusted the certainty level for PI to moderate based on GRADE guidance (see manuscript and Supplementary Table S2).

Reviewer #2: In Line 72, the manuscript mischaracterizes the PECO model as being based on interventions; however, PECO is more appropriately used for observational studies and is structured around exposure rather than intervention. This should be corrected, and an appropriate citation supporting the use of the PECO framework should be added.

The citation was added.

The study question is presented using the PECO format, but the individual components—Population, Exposure, Comparator, and Outcome—are not clearly defined. Each element should be explicitly stated to ensure clarity and methodological transparency.

The components were added.

The manuscript refers to the PRISMA guideline but does not provide a proper citation. A formal reference to the most current version of the PRISMA guideline (e.g., PRISMA 2020) should be included.

We added the citation.

The phrase “including papers, published and unpublished thesis” is imprecise. It should be revised to clearly describe the types of literature included in the review, specifying the nature of the unpublished materials and the sources used to retrieve them.

We revised the eligibility criteria section as requested.

The response to the earlier comment regarding the timing of the literature search is insufficient. The manuscript must explicitly indicate the date on which the literature search was conducted by your team for each database or platform. Merely referencing Line 83 is inadequate; the information should be clearly presented in the methods section.

As noted previously, the flow diagram in Figure 1 lacks specificity regarding exclusion criteria. The reasons for exclusion should be clearly stated both at the title and abstract screening stage and at the full-text review stage, in line with best practices for systematic review reporting.

The requested change was applied.

7. PLOS authors have the option to publish the peer review history of their article (what does this mean?). If published, this will include your full peer review and any attached files.

Do you want your identity to be public for this peer review? For information about this choice, including consent withdrawal, please see our Privacy Policy.

Reviewer #1: No

Reviewer #2: No

---

## [Editor Report · Decision Letter 2]

A Comparative Systematic Review and Meta-Analysis of Uterine Artery Resistance in Pregnant Women with and Without Previous History of Cesarean Section

PONE-D-24-48119R2

Dear Dr. Chaichian,

We’re pleased to inform you that your manuscript has been judged scientifically suitable for publication and will be formally accepted for publication once it meets all outstanding technical requirements.

Kind regards,

Mohammad Haddadi, MD

Academic Editor

PLOS ONE

Additional Editor Comments (optional):

Thank you for your efforts. Well written and all comments are addressed.
---

## [Editor Report · Acceptance letter]

PONE-D-24-48119R2

PLOS ONE

Dear Dr. Chaichian,

I'm pleased to inform you that your manuscript has been deemed suitable for publication in PLOS ONE. Congratulations! Your manuscript is now being handed over to our production team.

Kind regards,

on behalf of

Dr. Mohammad Haddadi

Academic Editor

PLOS ONE